# Streaming Min-Max Hypergraph Partitioning

**Dan Alistarh**
Microsoft Research
Cambridge, United Kingdom
dan.alistarh@microsoft.com

**Jennifer Iglesias**[*]
Carnegie Mellon University
Pittsburgh, PA
jiglesia@andrew.cmu.edu

**Milan Vojnovic**
Microsoft Research
Cambridge, United Kingdom
milanv@microsoft.com

## Abstract

In many applications, the data is of rich structure that can be represented by a hypergraph, where the data items are represented by vertices and the associations among items are represented by hyperedges. Equivalently, we are given an input bipartite graph with two types of vertices: items, and associations (which we refer to as topics). We consider the problem of partitioning the set of items into a given number of components such that the maximum number of topics covered by a component is minimized. This is a clustering problem with various applications, e.g. partitioning of a set of information objects such as documents, images, and videos, and load balancing in the context of modern computation platforms.

In this paper, we focus on the *streaming* computation model for this problem, in which items arrive online one at a time and each item must be assigned irrevocably to a component at its arrival time. Motivated by scalability requirements, we focus on the class of streaming computation algorithms with memory limited to be at most linear in the number of components. We show that a greedy assignment strategy is able to recover a hidden co-clustering of items under a natural set of recovery conditions. We also report results of an extensive empirical evaluation, which demonstrate that this greedy strategy yields superior performance when compared with alternative approaches.

## 1 Introduction

In a variety of applications, one needs to process data of rich structure that can be conveniently represented by a hypergraph, where associations of the data items, represented by vertices, are represented by hyperedges, i.e. subsets of items. Such data structure can be equivalently represented by a bipartite graph that has two types of vertices: vertices that represent *items*, and vertices that represent associations among items, which we refer to as *topics*. In this bipartite graph, each item is connected to one or more topics. The input can be seen as a graph with vertices belonging to (overlapping) communities.

There has been significant work on partitioning a set of items into disjoint components such that similar items are assigned to the same component, see, e.g., [8] for a survey. This problem arises in the context of *clustering of information objects* such as documents, images or videos. For example, the goal may be to partition given collection of documents into disjoint sub-collections such that the maximum number of distinct *topics* covered by each sub-collection is minimized, resulting in a

---

[*]Work performed in part while an intern with Microsoft Research.

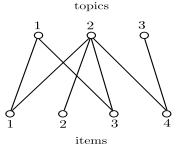

Figure 1: A simple example of a set of items with overlapping associations to topics.

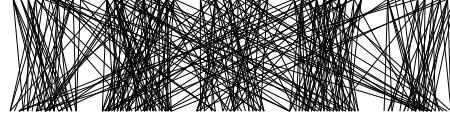

Figure 2: An example of hidden co-clustering with five hidden clusters.

parsimonious summary. The same fundamental problem also arises in *processing of complex data workloads*, including enterprise emails [10], online social networks [18], graph data processing and machine learning computation platforms [20, 21, 2], and load balancing in modern streaming query processing platforms [24]. In this context, the goal is to partition a set of data items over a given number of servers to balance the load according to some given criteria.

**Problem Definition.** We consider the min-max hypergraph partitioning problem defined as follows. The input to the problem is a set of *items*, a set of *topics*, a number of *components* to partition the set of items, and a *demand matrix* that specifies which particular subset of topics is associated with each individual item. Given a partitioning of the set of items, the cost of a component is defined as the *number of distinct topics* that are associated with items of the given component. The cost of a given partition is the *maximum* cost of a component. In other words, given an input hypergraph and a partition of the set of vertices into a given number of disjoints components, the cost of a component is defined to be the number of hyperedges that have at least one vertex assigned to this component. For example, for the simple input graph in Figure 1, a partition of the set of items into two components $\{1, 3\}$ and $\{2, 4\}$ amounts to the cost of the components each of value 2, thus, the cost of the partition is of value 2. The cost of a component is a submodular function as the distinct topics associated with items of the component correspond to a neighborhood set in the input bipartite graph.

In the streaming computation model that we consider, items arrive sequentially one at a time, and each item needs to be assigned, irrevocably, to one component at its arrival time. This streaming computation model allows for limited memory to be used at any time during the execution whose size is restricted to be at most linear in the number of the components. Both these assumptions arise as part of system requirements for deployment in web-scale services.

The min-max hypergraph partition problem is NP hard. The streaming computation problem is even more difficult, as less information is available to the algorithm when an item must be assigned.

**Contribution.** In this paper, we consider the streaming min-max hypergraph partitioning problem. We identify a greedy item placement strategy which outperforms all alternative approaches considered on real-world datasets, and can be proven to have a non-trivial recovery property: it recovers hidden co-clusters of items in probabilistic inputs subject to a recovery condition.

Specifically, we show that, given a set of hidden co-clusters to be placed onto $k$ components, the greedy strategy will tend to place items from the same hidden cluster onto the same component, with high probability. In turn, this property implies that greedy will provide a constant factor approximation of the optimal partition on inputs satisfying the recovery property.

The probabilistic input model we consider is defined as follows. The set of topics is assumed to be partitioned into a given number $\ell \geq 1$ of disjoint hidden clusters. Each item is connected to topics according to a mixture probability distribution defined as follows. Each item first selects one of the hidden clusters as a home hidden cluster by drawing an independent sample from a uniform distribution over the hidden clusters. Then, it connects to each topic from its home hidden cluster independently with probability $p$, and it connects to each topic from each other hidden cluster with probability $q \leq p$. This defines a hidden co-clustering of the input bipartite graph; see Figure 2 for an example.

This model is similar in spirit to the popular stochastic block model of an undirected graph, and it corresponds to a *hidden co-clustering* [6, 7, 17, 4] model of an undirected bipartite graph. We consider asymptotically accurate recovery of this hidden co-clustering.

A hidden cluster is said to be *asymptotically recovered* if the portion of items from the given hidden cluster assigned to the same partition goes to one asymptotically as the number of items observed grows large. An algorithm guarantees *balanced* asymptotic recovery if, additionally, it ensures that the cost of the most loaded partition is within a constant of the average partition load.

Our main analytical result is showing that a simple greedy strategy provides *balanced asymptotic recovery* of hidden clusters (Theorem 1). We prove that a sufficient condition for the recovery of hidden clusters is that the number of hidden clusters $\ell$ is at least $k \log k$, where $k$ is the number of components, and that the gap between the probability parameters $q$ and $p$ is sufficiently large: $q < \log r/(kr) < 2 \log r/r \leq p$, where $r$ is the number of topics in a hidden cluster. Roughly speaking, this means that if the mean number of topics to which an item is associated with in its home hidden cluster of topics is at least twice as large as the mean number of topics to which an item is associated with from other hidden clusters of topics, then the simple greedy online algorithm guarantees asymptotic recovery.

The proof is based on a coupling argument, where we first show that assigning an item to a partition based on the number of topics it has in common with each partition is similar to making the assignment proportionally to the number of *items* corresponding to the same hidden cluster present on each partition. In turn, this allows us to couple the assignment strategy with a *Polya urn process* [5] with "rich-get-richer" dynamics, which implies that the policy converges to assigning each item from a hidden cluster to the same partition. Additionally, this phenomenon occurs "in parallel" for each cluster. This recovery property will imply that this strategy will ensure a constant factor approximation of the optimum assignment.

Further, we provide experimental evidence that this greedy online algorithm exhibits good performance for several real-world input bipartite graphs, outperforming more complex assignment strategies, and even some offline approaches.

## 2 Problem Definition and Basic Results

In this section we provide a formal problem definition, and present some basic results on the computational hardness and lower bounds.

**Input.** The input is defined by a set of *items* $N = \{1, 2, \ldots, n\}$, a set of *topics* $M = \{1, 2, \ldots, m\}$, and a given number of components $k$. Dependencies between items and topics are given by a *demand matrix* $D = (d_{i,l}) \in \{0, 1\}^{n \times m}$ where $d_{i,l} = 1$ indicates that item $i$ needs topic $l$, and $d_{i,l} = 0$, otherwise.[1]

Alternatively, we can represent the input as a *bipartite graph* $G = (N, M, E)$ where there is an edge $(i, l) \in E$ if and only if item $i$ needs topic $l$ or as a *hypergraph* $H = (N, E)$ where a hyperedge $e \in E$ consists of all items that use the same topic.

**The Problem.** An assignment of items to components is given by $x \in \{0, 1\}^{n \times k}$ where $x_{i,j} = 1$ if item $i$ is assigned to component $j$, and $x_{i,j} = 0$, otherwise. Given an assignment of items to components $x$, the cost of component $j$ is defined to be equal to the minimum number of distinct topics that are needed by this component to cover all the items assigned to it, i.e.

$$c_j(x) = \sum_{l \in M} \min \left\{ \sum_{i \in N} d_{i,l} x_{i,j}, 1 \right\}.$$

As defined, the cost of each component is a *submodular function* of the items assigned to it. We consider the *min-max hypergraph partitioning problem* defined as follows:

$$
\begin{array}{ll}
\text{minimize} & \max\{c_1(x), c_2(x), \ldots, c_k(x)\} \\
\text{subject to} & \sum_{j \in [k]} x_{i,j} = 1 \quad \forall i \in [n] \\
& x \in \{0, 1\}^{n \times k}
\end{array}
\tag{1}
$$

We note that this problem is an instance of the submodular load balancing, as defined in [23].

**Basic Results.** This problem is NP-Complete, by reduction from the *subset sum* problem.

**Proposition 1.** *The min-max hypergraph partitioning problem is NP-Complete.*

We now give a lower bound on the optimal value of the problem, using the observation that each topic needs to be made available on at least one component.

**Proposition 2.** *For every partition of the set of items in $k$ components, the maximum cost of a component is larger than or equal to $m/k$, where $m$ is the number of topics.*

We next analyze the performance of an algorithm which simply assigns each item independently to a component chosen uniformly at random from the set of all components upon its arrival. Although this is a popular strategy commonly deployed in practice (e.g. for load balancing in computation platforms), the following result shows that it does not yield a good solution for the min-max hypergraph partitioning problem.

**Proposition 3.** *The expected maximum load of a component under random assignment is at least* $(1 - \sum_{j=1}^{m}(1 - 1/k)^{n_j}/m) \cdot m$, *where $n_j$ is the number of items associated with topic $j$.*

For instance, if we assume that $n_j \geq k$ for each topic $j$, we obtain that the expected maximum load is of at least $(1 - 1/e)m$. This suggests that the performance of random assignment is poor: on an input where $m$ topics form $k$ disjoint clusters, and each item subscribes to a single cluster, the optimal solution has cost $m/k$, whereas, by the above claim, random assignment has approximate cost $2m/3$, yielding a competitive ratio that is linear in $k$.

**Balanced Recovery of Hidden Co-Clusters.** We relax the worst-case input requirements by defining a family of *hidden co-clustering* inputs. Our model is a generalization of the stochastic block model of a graph to the case of hypergraphs.

We consider a set of topics $\mathcal{R}$, partitioned into $\ell$ clusters $C_1, C_2, \ldots, C_\ell$, each of which contains $r$ topics. Given these hidden clusters, each item is associated with topics as follows. Each item is first assigned a "home" cluster $C_h$, chosen uniformly at random among the hidden clusters. The item then connects to topics inside its home cluster by picking each topic independently with fixed probability $p$. Further, the item connects to topics from a fixed arbitrary "noise" set $Q_h$ of size at most $r/2$ outside its home cluster $C_h$, where the item is connected to each topic in $Q_h$ uniformly at random, with fixed probability $q$. (Sampling outside topics from the set of all possible topics would in the limit lead to every partition to contain *all* possible topics, which renders the problem trivial. We do not impose this limitation in the experimental validation.)

**Definition 1** (Hidden Co-Clustering). *A bipartite graph is in* $\mathsf{HC}(n, r, \ell, p, q)$ *if it is constructed using the above process, with $n$ items and $\ell$ clusters with $r$ topics per cluster, where each item subscribes to topics inside its randomly chosen home cluster with probability $p$, and to topics from the noise set with probability $q$.*

At each time step $t$, a new item is presented in the input stream of items, and is immediately assigned to one of the $k$ components, $S_1, S_2, \ldots, S_k$, according to some algorithm. Algorithms do not know the number of hidden clusters or their size, but can examine previous assignments.

**Definition 2** (Asymptotic Balanced Recovery.). *Given a hidden co-clustering* $\mathsf{HC}(n, r, \ell, p, q)$, *we say an algorithm* asymptotically recovers *the hidden clusters $C_1, C_2, \ldots, C_\ell$ if there exists a recovery time $t_R$ during its execution after which, for each hidden cluster $C_i$, there exists a component $S_j$ such that each item with home cluster $C_i$ is assigned to component $S_j$ with probability that goes to $1$ as the number of items grows large. Moreover, the recovery is* balanced *if the ratio between the* maximum cost of a component *and the* average cost over components *is upper bounded by a constant $B > 0$.*

## 3 Streaming Algorithm and the Recovery Guarantee

Recall that we consider the online problem, where we receive one item at a time together with all its corresponding topics. The item must be immediately and irrevocably assigned to some component. In the following, we describe the greedy strategy, specified in Algorithm 1.

**Data**: Hypergraph $H = (V, E)$, received one item (vertex) at a time, $k$ partitions, capacity bound $c$
**Result**: A partition of $V$ into $k$ parts
1 Set initial partitions $S_1, S_2, \ldots, S_k$ to be empty sets
2 **while** *there are incoming items* **do**
3   Receive the next item $t$, and its topics $R$
4   $I \leftarrow \{i : |S_i| \leq \min_j |S_j| + c\}$ /* components not exceeding capacity */
5   Compute $r_i = |S_i \cap R| \, \forall i \in I$ /* size of topic intersection */
6   $j \leftarrow \arg\max_{i \in I} r_i$ /* if tied, choose a least loaded component */
7   $S_j \leftarrow S_j \cup R$ /* item $t$ and its topics are assigned to $S_j$ */
8 return $S_1, S_2, \ldots, S_k$

**Algorithm 1:** The greedy algorithm.

This strategy places each incoming item onto the component whose incremental cost (after adding the item and its topics) is minimized. The immediate goal is not balancing, but rather clustering similar items. This could in theory lead to large imbalances; to prevent this, we add a *balancing constraint* specifying the maximum load imbalance. If adding the item to the first candidate component would violate the balancing constraint, then the item is assigned to the first valid component, in decreasing order of the intersection size.

## 3.1   The Recovery Theorem

In this section, we present our main theoretical result, which provides a sufficient condition for the greedy strategy to guarantee *balanced asymptotic recovery of hidden clusters*.

**Theorem 1** (The Recovery Theorem). *For a random input consisting of a hidden co-cluster graph $G$ in $\mathsf{HC}(n, r, \ell, p, q)$ to be partitioned across $k \geq 2$ components, if the number of clusters is $\ell \geq k \log k$, and the probabilities $p$ and $q$ satisfy $p \geq 2 \log r / r$, and $q \leq \log r / (rk)$, then the greedy algorithm ensures balanced asymptotic recovery of the hidden clusters.*

**Remarks.** Specifically, we prove that, under the given conditions, recovery occurs for each hidden cluster by the time $r / \log r$ cluster items have been observed, with probability $1 - 1/r^c$, where $c \geq 1$ is a constant. Moreover, clusters are randomly distributed among the $k$ components.

Together, these results can be used to bound the maximum cost of a partition to be at most a *constant* factor away the lower bound of $r\ell/k$ given by Lemma 2. The extra cost comes from incorrect assignments before the recovery time, and from the imperfect balancing of clusters over the components.

**Corollary 1.** *The expected maximum load of a component is at most $2.4 r\ell/k$.*

## 3.2   Proof Overview

We now provide an overview of the main ideas of the proof, which is available in the full version of the paper.

**Preliminaries.** We say that two random processes are *coupled* if their random choices are the same. We say that an event occurs *with high probability (w.h.p.)* if it occurs with probability at least $1 - 1/r^c$, where $c \geq 1$ is a constant. We make use of a Polya urn process [5], which is defined as follows. We start each of $k \geq 2$ urns with one ball, and, at each step $t$, observe a new ball. We assign the new ball to urn $i \in \{1, \ldots, k\}$ with probability proportional to $(b_i)^\gamma$, where $\gamma > 0$ is a fixed real constant, and $b_i$ is the number of balls in urn $i$ at time $t$. We use the following classic result.

**Lemma 1** (Polya Urn Convergence [5]). *Consider a finite $k$-bin Polya urn process with exponent $\gamma > 1$, and let $x_i^t$ be the fraction of balls in urn $i$ at time $t$. Then, almost surely, the limit $X_i = \lim_{t \to \infty} x_i^t$ exists for each $1 \leq i \leq k$. Moreover, we have that there exists an urn $j$ such that $X_j = 1$, and that $X_i = 0$, for all $i \neq j$.*

**Step 1: Recovering a Single Cluster.** We first prove that, in the case of a single home cluster for all items, and two components ($k = 2$), with no balance constraints, the greedy algorithm with no balance constraints converges to a *monopoly*, i.e., eventually assigns all the items from

| Dataset | Items | Topics | # of Items | # of Topics | # edges |
|---------|-------|--------|-----------|------------|---------|
| Book Ratings | Readers | Books | 107,549 | 105,283 | 965,949 |
| Facebook App Data | Users | Apps | 173,502 | 13,604 | 5,115,433 |
| Retail Data | Customers | Items bought | 74,333 | 16,470 | 947,940 |
| Zune Podcast Data | Listeners | Podcasts | 80,633 | 7928 | 1,037,999 |

Figure 3: A table showing the data sets and information about the items and topics.

this cluster onto the same component, w.h.p. Formally, there exists some convergence time $t_R$ and some component $S_i$ such that, after time $t_R$, all future items will be assigned to component $S_i$, with probability at least $1 - 1/r^c$.

Our strategy will be to couple greedy assignment with a Polya urn process with exponent $\gamma > 1$, showing that the dynamics of the two processes are the same, w.h.p. There is one significant technical challenge that one needs to address: while the Polya process assigns new *balls* based on the *ball* counts of urns, greedy assigns *items* (and their respective topics) based on the number of *topic intersections* between the item and the partition. We resolve this issue by taking a two-tiered approach. Roughly, we first prove that, w.h.p., we can couple the number of items in a component with the number of *unique topics* assigned to the same component. We then prove that this is enough to couple the greedy assignment with a Polya urn process with exponent $\gamma > 1$. This will imply that greedy converges to a monopoly, by Lemma 1.

We then extend this argument to a single cluster and $k \geq 3$ components, but with no load balancing constraints. The crux of the extension is that we can apply the $k = 2$ argument to *pairs of components* to yield that some component achieves a monopoly.

**Lemma 2.** *Given a single cluster instance in* $\mathsf{HC}(n, r, \ell, p, q)$ *with* $\ell = 1$, $p \geq 2\log r/r$ *and* $q = 0$ *to be partitioned in* $k$ *components, the greedy algorithm with no balancing constraints will eventually place every item in the cluster onto the same component w.h.p.*

**Second Step: The General Case.** We complete the proof of Theorem 1 by considering the general case with $\ell \geq 2$ clusters and $q > 0$. We proceed in three sub-steps. We first show the recovery claim for general number of clusters $\ell \geq 2$, but $q = 0$ and no balance constraints. This follows since, for $q = 0$, the algorithm's choices with respect to clusters and their respective topics are independent. Hence clusters are assigned to components *uniformly at random*.

Second, we extend the proof for any value $q \leq \log r/(rk)$, by showing that the existence of "noise" edges under this threshold only affects the algorithm's choices with very low probability. Finally, we prove that the balance constraints are practically never violated for this type of input, as clusters are distributed uniformly at random. We obtain the following.

**Lemma 3.** *For a hidden co-cluster input, the greedy algorithm with* $q = 0$ *and without capacity constraints can be coupled with a version of the algorithm with* $q \leq \log r/(rk)$ *and a constant capacity constraint, w.h.p.*

**Final Argument.** Putting together Lemmas 2 and 3, we obtain that greedy ensures balanced recovery for general inputs in $\mathsf{HC}(n, r, \ell, p, q)$, for parameter values $\ell \geq k\log k$, $p \geq 2\log r/r$, and $q \leq \log r/(rk)$.

## 4 Experimental Results

**Datasets and Evaluation.** We first consider a set of real-world bipartite graph instances with a summary provided in Table 3. All these datasets are available online, except for Zune podcast subscriptions. We chose the consumer to be the item and the resource to be the topic. We provide an experimental validation of the analysis on synthetic co-cluster inputs in the full version of our paper.

In our experiments, we considered partitioning of items onto $k$ components for a range of values going from two to ten components. We report the maximum number of topics in a component normalized by the cost of a perfectly balanced solution $m/k$, where $m$ is the total number of topics.

**Online Assignment Algorithms.** We compared the following other online assignment strategies:

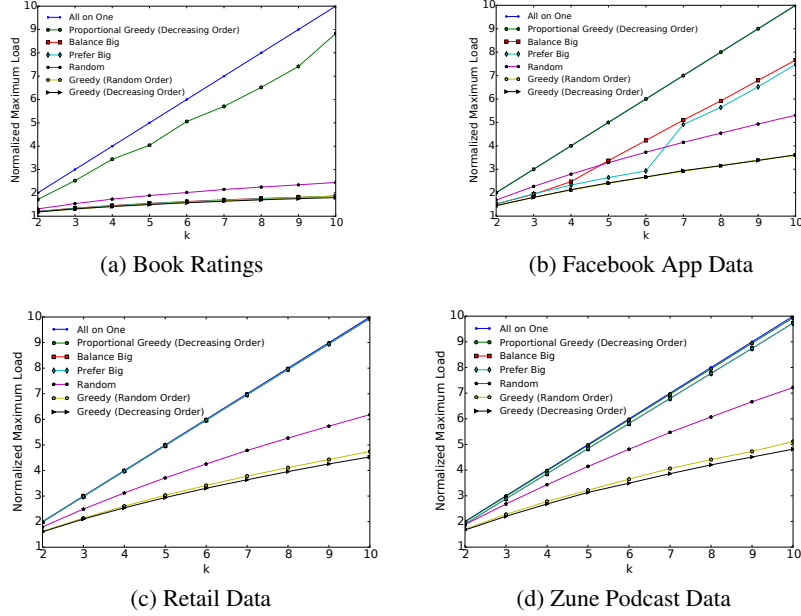

(a) Book Ratings  (b) Facebook App Data

(c) Retail Data  (d) Zune Podcast Data

Figure 4: The normalized maximum load for various online assignment algorithms under different input bipartite graphs versus the numbers of components.

- *All-on-One*: trivially assign all items and topics to one component.

- *Random*: assign each item independently to a component chosen uniformly at random from the set of all components.

- *Balance Big*: inspect the items in a random order and assign the large items to the least loaded component, and the small items according to greedy. An item is considered large if it subscribes to more than 100 topics, and small otherwise.

- *Prefer Big*: inspect the items in a random order, and keep a buffer of up to 100 small items; when receiving a large item, put it on the least loaded component; when the buffer is full, place all the small items according to greedy.

- *Greedy*: assign the items to the component they have the most topics in common with. We consider two variants: items arrive in *random order*, and items arrive in *decreasing order* of the number of topics. We allow a slack (parameter $c$) of up to 100 topics.

- *Proportional Allocation*: inspect the items in decreasing order of the number of topics; the probability an item is assigned to a component is proportional to the number of common topics.

**Results.** Greedy generally outperforms other online heuristics (see Figure 4). Also, its performance is improved if items arrive in decreasing order of number of topics. Intuitively, items with larger number of topics provide more information about the underlying structure of the bipartite graph than the items with smaller number of topics. Interestingly, adding randomness to the greedy assignment made it perform far worse; most times Proportional Assignment approached the worst case scenario. Random assignment outperformed Proportional Assignment and regularly outperformed Prefer Big and Balance Big item assignment strategies.

**Offline methods.** We also tested the streaming algorithm for a wide range of synthetic input bipartite graphs according to the model defined in this paper, and several offline approaches for the problem including hMetis [11], label propagation, basic spectral methods, and PARSA [13]. We found that label propagation and spectral methods are extremely time and memory intensive on our inputs, due to the large number of topics and item-topic edges. hMetis returns within seconds, however the assignments were not competitive. However, hMetis provides balanced hypergraph cuts, which are not necessarily a good solution to our problem.

Compared to PARSA on bipartite graph inputs, greedy provides assignments with up to 3x higher max partition load. On social graphs, the performance difference can be as high as 5x. This discrepancy is natural since PARSA has the advantage of performing multiple passes through the input.

## 5   Related Work

The related problem of min-max multi-way graph cut problem, originally introduced in [23], is defined as follows: given an input graph, the objective is to component the set of vertices such that the maximum *number of edges* adjacent to a component is minimized. A similar problem was recently studied, e.g. [1], with respect to *expansion*, defined as the ratio of the sum of weights of edges adjacent to a component and the minimum between the sum of the weights of vertices within and outside the given component. The *balanced graph partition* problem is a bi-criteria optimization problem where the goal is to find a balanced partition of the set of vertices that minimizes the total number of edges cut. The best known approximation ratio for this problem is poly-logarithmic in the number of vertices [12]. The balanced graph partition problem was also considered for the set of *edges* of a graph [2]. The related problem of community detection in an input graph data has been commonly studied for the *planted partition model*, also well known as *stochastic block model*. Tight conditions for recovery of hidden clusters are known from the recent work in [16] and [14], as well as various approximation algorithms, e.g. see [3]. Some variants of hypergraph partition problems were studied by the machine learning research community, including balanced cuts studied by [9] using relaxations based on the concept of total variation, and the maximum likelihood identification of hidden clusters [17]. The difference is that we consider the min-max multi-way cut problem for a hypergraph in the streaming computation model. PARSA [13] considers the same problem in an offline model, where the entire input is initially available to the algorithm, and provides an efficient distributed algorithm for optimizing multiple criteria. A key component of PARSA is a procedure for optimizing the order of examining vertices. By contrast, we focus on performance under arbitrary arrival order, and provide analytic guarantees under a stochastic input model.

Streaming computation with limited memory was considered for various canonical problems such as principal component analysis [15], community detection [22], balanced graph partition [20, 21], and query placement [24]. For the class of (hyper)graph partition problems, most of the work is restricted to studying various streaming heuristics using empirical evaluations with a few notable exceptions. A first theoretical analysis of streaming algorithms for balanced graph partitioning was presented in [19] using the framework similar to the one deployed in this paper. The paper gives sufficient conditions for a greedy streaming strategy to recover clusters of vertices for the input graph according to stochastic block model, which makes irrevocable assignments of vertices as they are observed in the input stream and uses memory limited to grow linearly with the number of clusters. As in our case, the argument uses a reduction to Polya urn processes. The two main differences with our work is that we consider a different problem (min-max hypergraph partition) and this requires a novel proof technique based on a two-step reduction to Polya urn processes. Streaming algorithms for the recovery of clusters in a stochastic block model were also studied in [22], under a weaker computation model, which does not require irrevocable assignments of vertices at instances they are presented in the input stream and allows for memory polynomial in the number of vertices.

## 6   Conclusion

We studied the min-max hypergraph partitioning problem in the streaming computation model with the size of memory limited to be at most linear in the number of the components of the partition. We established first approximation guarantees for inputs according to a random bipartite graph with hidden co-clusters, and evaluated performance on several real-world input graphs. There are several interesting open questions for future work. It is of interest to study the tightness of the given recovery condition, and, in general, better understand the trade-off between the memory size and the accuracy of the recovery. It is also of interest to consider the recovery problem for a wider set of random bipartite graph models. Another question of interest is to consider dynamic graph inputs with addition and deletion of items and topics.

## Footnotes

[1]The framework allows for a natural generalization to allow for real-valued demands. In this paper we focus on $\{0, 1\}$-valued demands.

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
