[Supplementary Material · additional.pdf]

# Additional Material for
# Streaming Min-max Hypergraph Partitioning

**Dan Alistarh**
Microsoft Research
Cambridge, United Kingdom
dan.alistarh@microsoft.com

**Jennifer Iglesias**[*]
Carnegie Mellon University
Pittsburgh, PA
jiglesia@andrew.cmu.edu

**Milan Vojnovic**
Microsoft Research
Cambridge, United Kingdom
milanv@microsoft.com

**Outline.** In this document, we provide the complete proof of the recovery theorem (Section 1), as well as experimental validation of the argument (Section 2).

## 1  Complete Proof of the Recovery Theorem

In this section, we show that the greedy strategy is expected to perform well on a *hidden co-cluster* input.

### 1.1  Balanced Recovery of Hidden Co-Clusters

We recall the definition of our hidden co-cluster model. This model is a generalization of the graph Stochastic Block Model to the case of hypergraphs.

**Generative Process.** We consider a set of topics $\mathcal{R}$, partitioned into $\ell$ clusters $C_1, C_2, \ldots, C_\ell$, each of which contains $r$ topics. Based on these hidden clusters, an incoming item $\tau$ is associated with topics as follows. The item $\tau$ is first assigned to a "home" cluster $C_h$, chosen uniformly at random among the hidden clusters. The item then subscribes to topics inside its cluster, picking each topic independently with some probability $p$. Further, the item subscribes to topics from a fixed arbitrary "noise" set $Q_h$ of size $\leq r/2$ outside its home cluster $C_h$, where each topic in $Q_h$ is subscribed to uniformly at random, with some probability $q$.[1]

**Definition 1** (Hidden Co-Clustering). *A bipartite graph is in* $\mathsf{HC}(n, r, \ell, p, q)$ *if it is constructed using the above process, with $n$ items and $\ell$ clusters with $r$ topics per cluster, where each item subscribes to topics inside its randomly chosen home cluster with probability $p$, and to topics from the noise set with probability $q$.*

**Online Allocation of Items.** We iterate the generative process over time, where at each time step $t$ we generate a new item, and consider a set of $k$ partitions $S_1, S_2, \ldots, S_k$. At each time step, the incoming item is immediately assigned to one of the $k$ partitions, together with all its topics, according to some algorithm. Algorithms do not know the number of hidden clusters or their size, but can examine previous assignments.

---

[*]Work performed in part while an intern with Microsoft Research.

[1]Sampling outside topics from the set of all possible topics would eventually lead every partition to contain *all* possible topics, which renders the problem trivial. However, we do not impose this limitation in the experimental validation of our analysis.

```
    Data: Hypergraph H = (V, E), received one item (vertex) at a time, k partitions, capacity bound c
    Result: A partition of V into k parts
1  Set initial partition assignments S₁, S₂, ..., Sₖ to be empty sets
2  while there are incoming items do
3      Receive the next item t, and its topics R
4      I ← {i : |Sᵢ| ≤ minⱼ |Sⱼ| + c} /* partitions not exceeding capacity */
5      Compute rᵢ = |Sᵢ ∩ R| ∀i ∈ I /* size of topic intersection  */
6      j ← arg maxᵢ∈I rᵢ /* if tied, choose least loaded partition  */
7      Sⱼ ← Sⱼ ∪ R /* item t has been assigned to partition Sⱼ  */
8  return S₁, S₂, ..., Sₖ
```

**Algorithm 1:** The greedy algorithm.

**Definition 2** (Asymptotic Balanced Recovery.)**.** *Given a hidden co-clustering* $\mathsf{HC}(n, r, \ell, p, q)$, *we say an algorithm* asymptotically recovers *the hidden clusters* $C_1, C_2, \ldots, C_\ell$ *if there exists a recovery time* $t_R$ *during its execution after which, for each hidden cluster* $C_i$, *there exists a partition* $S_j$ *such that each item with home cluster* $C_i$ *is assigned to partition* $S_j$ *with probability that goes to* $1$.

*Moreover, recovery is* balanced *if the ratio between the* maximum partition cost *and the* average partition cost *is upper bounded by a constant* $B > 0$.

## 1.2  The Greedy Online Algorithm

We recall the structure of the greedy strategy, described in Algorithm 1.

## 1.3  Theorem Statement and Proof Outline

Our main technical result provides sufficient conditions on the cluster parameters for the greedy strategy to provide *balanced recovery of hidden clusters, with high probability.*

**Theorem 1** (The Recovery Theorem)**.** *For a random input consisting of a hidden co-cluster graph* $G$ *in* $\mathsf{HC}(n, r, \ell, p, q)$ *to be distributed across* $k \geq 2$ *partitions, if the number of clusters is* $\ell \geq k \log k$, *and the probabilities* $p$ *and* $q$ *satisfy* $p \geq 2 \log r / r$, *and* $q \leq \log r / (rk)$, *then greedy ensures balanced asymptotic recovery of the hidden clusters.*

**Coupling and High Probability.** In the following, we say that two random processes are *coupled* to mean that their random choices are the same. We say that an event occurs *with high probability (w.h.p.)* if it occurs with probability at least $1 - 1/r^c$, where $c \geq 1$ is a constant.

**Proof Overview.** The proof of this result can be summarized as follows. The first step will be to prove that greedy recovers a single cluster w.h.p. when assigning to just two partitions. More precisely, given a sequence of items generated from a single home cluster, and two partitions, a version of the algorithm without balancing constraints will eventually converge to assigning all incoming items to a single partition. This is a main technical step of the proof, and it is based on a coupling of greedy assignment with a "rich get richer" *Polya urn process* [1], and then using the convergence properties of such processes. Further, we extend this coupling claim from two partitions to $k > 2$ partitions, again for a single cluster, showing that, when the input consists of items from a single cluster, greedy will quickly converge to assigning all items to a single partition, w.h.p.

In the next step, we prove that the algorithm will in fact recover $\ell$ clusters of items in parallel, assigning each of them (i.e., most of their corresponding items) independently at random to one of the partitions, and that this convergence is not adversely affected by the fact that items also subscribe to topics from outside their home cluster. The problem of determining the maximum partition load is then reduced to showing that the maximum number of clusters that may be randomly assigned to a partition is *balanced*, as well as bounding the extra load due on a server to topics outside the home cluster and miss-assignments.

**Polya Urn Processes.** For reference, a Polya urn process [1] works as follows. We start each of $k \geq 2$ urns with one ball, and, at each step $t$, observe a new ball. We assign the new ball to urn

$i \in \{1, \ldots, k\}$ with probability proportional to $(b_i)^\gamma$, where $\gamma > 0$ is a fixed real constant, and $b_i$ is the number of balls in urn $i$ at time $t$. We shall employ the following classic result.

**Lemma 1** (Polya Urn Convergence [1]). *Consider a finite $k$-bin Polya urn process with exponent $\gamma > 1$, and let $x_i^t$ be the fraction of balls in urn $i$ at time $t$. Then, almost surely, the limit $X_i = \lim_{t \to \infty} x_i^t$ exists for each $1 \le i \le k$. Moreover, we have that there exists an urn $j$ such that $X_j = 1$, and that $X_i = 0$, for all $i \ne j$.*

### 1.4 Step 1: Recovering a Single Cluster

**Strategy.** We first prove that, in the case of a single home cluster for all items, and two partitions ($k = 2$), with no balance constraints, the greedy algorithm with no balance constraints converges to a *monopoly*, i.e., eventually assigns all the items from the cluster onto the same partition, w.h.p. Formally, there exists some convergence time $t_R$ and some partition $S_i$ such that, after time $t_R$, all future items associated to this home cluster will be assigned to partition $S_i$, with probability at least $1 - 1/r^c$.

Our strategy will be to couple greedy assignment with a Polya urn process with exponent $\gamma > 1$, showing that the dynamics of the two processes are the same, w.h.p. There is one serious technical issue: while the Polya process assigns new *balls* based on the *ball* counts of urns, greedy assigns *items* (and their respective topics) based on the number of *topic intersections* between the item and the partition. It is not clear how these two measures are related.

We circumvent this issue by taking a two-tiered approach. Roughly, we first prove that, w.h.p., we can couple the number of items on a server with the number of *unique topics* assigned to the same partition. We then prove that this is enough to couple the greedy assignment with a Polya urn process with exponent $\gamma > 1$ (Lemma 4). This will imply that greedy converges to a monopoly, by Lemma 1.

**Notation.** Fix a time $t$ in the execution of the greedy assignment process, corresponding to some new item being randomly generated. A topic $r$ is *known* at time $t$ if it has been a topic for some item up to time $t$. A known topic $r$ is a *singleton* if it has been placed on one partition, but not on others. Otherwise, it is a *duplicate*. In the following, we will focus on the above quantities around the special time $t_0 = r / \log r$, which we shall prove is close to the convergence time. For simplicity, when referring to a value at time $t_0$, we omit the subscript.

#### 1.4.1 Auxiliary Results

We first state some helper results characterizing the number of known and singleton topics up to some point in time. The reader may skip this sub-section, and return to it as necessary while reading the proof of Lemma 4.

**Lemma 2.** *The following hold.*

1. *For $0 < \epsilon < 1$ constant, the number of topics inside the cluster to which an item $\tau$ subscribes is in $[2(1-\epsilon) \log r, 2(1+\epsilon) \log r]$, w.h.p.*

2. *The expected number of known topics by time $t$ is at least $r(1 - \exp(-2t \log r / r))$.*

3. *For any time $t \ge r / \log^2 r$, the number of known topics is at least $r / \log r$ and the number of* singleton *topics is at least $r/(2 \log r)$, both w.h.p.*

*Proof.* The first statement follows by straightforward application of Chernoff bounds. To bound the number of known topics, notice that, at each step, a specific topic $r$ is sampled with probability $p$. Therefore, the probability that $r$ has not been sampled by time $t$ is $(1-p)^t$. Plugging in $p = 2 \log r / r$, it follows that the expected number of unknown topics up to $t$ is $r \exp(-2t \log r / r)$, which implies the second claim.

In particular, this number of unknown topics by time $t = r / \log^2 r$ is at most $r / e^{2/\log r} \le r(1 - 3/2 \log r)$, by the Taylor expansion. Therefore, the expected number of known topics up to $t$ is at least $3r/(2 \log r)$. By a Chernoff bound, it follows that the number of known topics up to $t$ is at most $r / \log r$, w.h.p., as required.

To lower bound the number of singleton topics, notice that it is sufficient to lower bound the number of topics that have been sampled exactly once up to and including time $t$. (Such topics are necessarily singletons.) The probability that a topic has been sampled *exactly once* is $tp(1-p)^{t-1}$. Since $t \geq r/\log^2 r$, we obtain that the expected number of topics that have been sampled exactly once is at least $r/(\log re^{1/\log r}) \geq r/\log r$, for large enough $r$. Hence, the number of singletons up to this point is at least $r/(2\log r)$, w.h.p., which completes the proof. $\qquad\square$

We next focus on the ratio of singleton topics between the two partitions. Define the *singleton topic ratio* as the number of singleton topics on the more loaded partition divided by the number of singleton topics on the less loaded partition. Further, define the *topic-to-item* quotient of a partition as the number of topics it contains divided by the number of items that have been assigned to it.

**Lemma 3.** *Assume that the singleton topic ratio at time $t_0$ is $\mu \leq 1/2 + \epsilon$, for fixed $\epsilon < 1$, and let $\phi(\epsilon) = \left(\frac{1/2+2\epsilon}{1/2-2\epsilon}\right)^2$. Then, the ratio between the topic-to-item quotients of the two partitions is in the interval $[1/\phi(\epsilon), \phi(\epsilon)]$, with high probability.*

*Proof.* Let $\sigma_i$ denote the number of singleton topics on partition $i$. Without loss of generality, let partition 1 be the more loaded one at $t_0$, i.e., $\mu = \sigma_1/(\sigma_1 + \sigma_2)$. Let $T_1$ be the set of items assigned to the first partition between times $r/\log^2 r$ and $r/\log r$, and $T_2$ be the corresponding set of items for partition 2. By the Lemma statement, we have that $\sigma_1/(\sigma_1 + \sigma_2) \leq 1/2 + \epsilon$ at $t_0$.

Given this bound, our first claim is as follows. If $\mu \leq 1/2+\epsilon$ at time $t_0$, then, for all times $r/\log^2 r \leq t \leq r/\log r$, we have that $\mu_t \leq 1/2 + 2\epsilon$, w.h.p. Also, $|T_1|/(|T_1| + |T_2|) \in [1/2 - 2\epsilon, 1/2 + \epsilon]$, w.h.p.

We focus on the proof of the first statement above, and the second will follow as a corollary. Let us assume for contradiction the converse, i.e., that there exists a time step $r/\log^2 r \leq t \leq r/\log r$ for which $\mu_t > 1/2 + 2\epsilon$. We will show that, after time $t$, the relative gap in terms of singleton topics between the two partitions will increase, with high probability, which contradicts the bound at time $t_0$.

For this, consider an incoming item $\tau$. If this item is subscribing to a known topic, which we call case 1, then it will be assigned by the intersection rule. In case 2, it will be placed uniformly at random on one of the partitions. To control this process, we split the execution from time $t$ into blocks of $b = r/\log^4 r$ consecutive incoming items. Notice that, by Lemma 2, there are at least $r/\log r$ known topics after time $r/\log^2 r$, w.h.p. This implies that the probability that an item is *not* assigned by the intersection rule after this time is at most $(1 - 2\log r/r)^{r/\log r} \leq (1/e)^2$. Therefore, each incoming item is assigned by the intersection rule after this time, with at least constant probability.

Consider now the probability that a case-1 item gets assigned to partition 1, assuming that $\sigma_1/(\sigma_1 + \sigma_2) > 1/2 + 2\epsilon$ at the beginning of the current block. This means that the item has more topics in common with partition 1 than with partition 2. By calculation, this probability is at least

$$\frac{1/2 + 2\epsilon}{1/2 - 2\epsilon + 7b\log r/r + 1/2 + 2\epsilon} \geq 1/2 + 7\epsilon/4,$$

where we have pessimistically assumed that *all the items* in the current block get assigned to the second partition, and that each such item contains at most $7/3 \log r$ new topics. (This last fact holds w.h.p. by Lemma 2.)

For an item $i$ during this block, let $X_i$ be an indicator random variable for the event that the item gets assigned to partition 1, and fix $X = \sum_i X_i$. We wish to lower bound $X$, and will assume that these events are independent—the fact that they are positively correlated does not affect the lower bound. We apply Chernoff bounds, to get that, w.h.p., $X \geq (1 - \delta)7b\epsilon/4$, that is, the first partition gets at least $(1 - \delta)7b\epsilon/4$ *extra* items from each block, where $0 < \delta < 1$ is a constant. On the other hand, the number of case-2 items assigned is balanced, since these items are assigned randomly. In particular, it can be biased towards partition 2 by a fraction of at most $(1 - \delta)\epsilon/4$, w.h.p. We have therefore obtained that partition 1 obtains an *extra* number of items which is at least $3b\epsilon/2$ in each block, w.h.p. Summing over $\log^2 r$ blocks, we get that, over a period of $r/\log^2 r$ time steps, partition 1 gets at least $(1/2 + 3\epsilon/2)r/\log^2 r$ extra items, w.h.p.

Notice that, in turn, this item advantage also translates into a similar extra proportion of *new topics* acquired during each block. In particular, we obtain that the first partition acquires an $(1/2 + 4\epsilon/3)$ fraction of the new topics observed in a block, w.h.p. Recall that, by assumption, at the beginning of the process, partition 1 already had a fraction of $(1/2 + 2\epsilon)$ singleton topics. Therefore, the event that the singleton topic ratio is balanced by at most $1/2 + \epsilon$ at $t_0$ has very low probability, as claimed. The proof of the second statement follows by a similar argument.

To complete the proof of Lemma 3, it is enough to notice that, by the previous claim, the ratio between the topic-to-item quotients of the two partitions is bounded as $\frac{\sigma_1 + \kappa}{\sigma_2 + \kappa} \cdot \frac{q_2}{q_1} \leq \left( \frac{1/2 + 2\epsilon}{1/2 - 2\epsilon} \right)^2$, which completes the proof of Lemma 3. $\qquad\square$

### 1.4.2 Convergence to a Monopoly

We can now prove that one of two things must happen during the algorithm's execution: either one of the partitions gains a constant size advantage, or the algorithm can be coupled with a Polya urn process. In both cases, the algorithm will converge to a monopoly.

**Lemma 4.** *Given a hidden cluster input* $\mathsf{HC}(n, r, \ell, p, q)$, *with* $\ell = 1$, $p \geq 2 \log r / r$ *and* $q = 0$, *for every* $t \geq t_0 = r / \log r$, *to be allocated onto two partitions, one of the following holds:*

1. *With high probability, the greedy algorithm with a cluster and two partitions can be coupled with a finite Polya urn process with parameter* $\gamma > 1$, *or*

2. *There exists a constant* $\rho > 0$ *such that the ratio between the number of singleton topics on the two partitions is* $> 1 + \rho$ *at time* $t_0$.

*Further, in both cases, the algorithm converges to assigning all incoming items to a single partition after some time* $t = O(r / \log r)$, *w.h.p.*

*Proof.* We proceed by induction on the time $t \geq t_0$. We will focus on time $t_0 = r / \log r$, as the argument is similar for larger values of $t$. Notice that we have two cases at $t_0$. If there exists a constant $\rho > 0$ such that the ratio between the number of singleton topics on the two partitions is $> 1 + \rho$ at time $t$, then we are obviously done by case 2.

Therefore, in the following, we will work in the case where the load ratio between the two partitions at time $t_0$ is $\leq 1 + \rho$. Without loss of generality, assume $1 \leq (\sigma_1 + \kappa)/(\sigma_2 + \kappa) \leq 1 + \rho$.

By Lemma 2, the number of singleton topics at time $t \geq t_0$ is at least a constant fraction of $r$, w.h.p., and it follows that there exists a constant $\epsilon > 0$ such that the singleton ratio at time $t_0$ is at most $1 + \epsilon$. Also, the probability that an item with $3 \log r / 2$ distinct topics does not hit any of these known topics is at most $1/r^{3/2}$. Hence, in the following, we can assume w.h.p. that every incoming item is assigned by the intersection rule.

By Lemma 3, the ratio between the topic-to-item quotients of the two partitions at time $t_0$ is at most $\left( \frac{1/2 + 2\epsilon}{1/2 - 2\epsilon} \right)^2$, w.h.p. We now proceed to prove that in this case the greedy assignment process can be coupled with a Polya urn process with $\gamma > 1$, w.h.p., noting that this part of the proof is similar to the coupling argument in [3].

By Lemma 2, for $t \geq r / \log r$ steps, at least $2r/3$ topics have been observed, w.h.p. Therefore, the probability that an item with $3 \log r / 2$ topics does not hit any of these known topics is at most $1/r^{3/2}$. Hence, in the following, we can safely assume that every incoming item is assigned by the intersection rule.

More precisely, when a new item comes in, we check the intersection with the number of topics on each server, and assign it to the partition with which the intersection is larger. (Or randomly if the intersections are equal.) Given an item $\tau$ observed at time $t \geq r / \log r$, let $A$ be the number of topics it has in common with partition 1, and $B$ be the number it has in common with partition 2.

More precisely, fix $j \geq 0$ to be the size of the total intersection with either partition, and let $a$ and $b$ be the values of the intersections with partitions 1 and 2, respectively, conditioned on the fact that $a + b = j$. Let $\delta$ be the advantage in terms of *topics* of partition 1 versus partition 2, i.e.

$(\sigma_1 + \kappa)/(\sigma_1 + \sigma_2 + 2\kappa) = 1/2 + \delta$, and $(\sigma_2 + \kappa)/(\sigma_1 + \sigma_2 + 2\kappa) = 1/2 - \delta$, where $\kappa$ is the number of duplicate topics. We now analyze the probability that $a > b$.

We can see this as a one-dimensional random walk, in which we start at $0$, and take $j$ steps, going right with probability $(1/2+\delta)$, and left with probability $(1/2-\delta)$. We wish to know the probability that we have finished to the right of $0$. Iterating over $i$, the possible value of our drift to the right, we have that

$$\Pr[a > b] = \sum_{i=[j/2]+1}^{j} \binom{j}{i} \left(\frac{1}{2} + \delta\right)^i \left(\frac{1}{2} - \delta\right)^{j-i} =$$

$$\left(\frac{1}{2} + \delta\right)^{[j/2]+1} \sum_{i=0}^{[j/2]} \binom{j}{i} \left(\frac{1}{2} + \delta\right)^{[j/2]-i} \left(\frac{1}{2} - \delta\right)^i .$$

Similarly, we obtain that

$$\Pr[a < b] = \left(\frac{1}{2} - \delta\right)^{[j/2]+1} \sum_{i=0}^{[j/2]} \binom{j}{i} \left(\frac{1}{2} + \delta\right)^i \left(\frac{1}{2} - \delta\right)^{[j/2]-i} .$$

Since $\delta > 0$, we have that the sum on the right-hand-side of the first equation dominates the term on the right-hand-side of the second equation. It follows that

$$\frac{\Pr[a > b]}{\Pr[a < b]} > \frac{\left(\frac{1}{2} + \delta\right)^{[j/2]+1}}{\left(\frac{1}{2} - \delta\right)^{[j/2]+1}} .$$

Since the two quantities sum up to (almost) $1$, we obtain that

$$\Pr[a > b] > \frac{\left(\frac{1}{2} + \delta\right)^{[j/2]+1}}{\left(\frac{1}{2} + \delta\right)^{[j/2]+1} + \left(\frac{1}{2} - \delta\right)^{[j/2]+1}} .$$

Let $\delta'$ be the advantage that the first partition has over the second *in terms of number of items*, i.e. $1/2 + \delta' = q_1/(q_1 + q_2)$. Using Lemma 3, and setting $\epsilon$ to a small constant, we obtain that $\delta \simeq \delta'$. We can therefore express the same lower bound in terms of $\delta'$.

$$\Pr[a > b] > \frac{\left(\frac{1}{2} + \delta'\right)^{[j/2]+1}}{\left(\frac{1}{2} + \delta'\right)^{[j/2]+1} + \left(\frac{1}{2} - \delta'\right)^{[j/2]+1}} .$$

The lower bound on the right-hand-side is the probability that the ball goes in urn $1$ in a Polya process with $\gamma = [j/2] + 1$. Importantly, notice that, in this process, we are assigning balls (items) with probability proportional to the number of *balls (items)* present in each bin, and have thus eliminated topics from the choice. Let $\beta_t$ be the proportion of singletons at time $t$, i.e. $(\sigma_1 + \sigma_2)/r$. We can then eliminate the conditioning on $j$ to obtain that

$$\Pr[A > B] \geq \sum_{j=1}^{d} \binom{d}{j} (\beta_t/r)^j (1 - \beta_t/r)^{d-j} \Pr[a > b|j]. \tag{1}$$

The only case where greedy is coupled with a Polya urn process with undesirable exponent $\gamma \leq 1$ is when $j \leq 1$. However, since an item has at least $3 \log r/2$ distinct topics, w.h.p., and $t \geq t_0 = r/\log r$, the probability that we hit $j \leq 1$ topics is negligible. Therefore we can indeed couple our process to a finite Polya urn process with $\gamma > 1$ at time $t_0$ in the case where the singleton ratio at $t_0$ is at most $1/2 + \epsilon$, for $\epsilon$ a small constant. We can apply the same argument by induction for all times $t \geq t_0$, noticing that, once the load ratio is larger than a fixed constant, it never falls below that constant, except with low probability. $\qquad\square$

### 1.5 Step 2: $k$ Partitions and Convergence

**Multiple Partitions.** Consider now greedy on $k \geq 3$ partitions, but with no load balancing constraint. We now extend the previous argument to this case.

Let $t \geq r / \log r$, and consider the state of the partitions at time $t$. If there exists a set of partitions which have a constant fraction more singleton topics than the others, it follows by a simple extension of Lemma 4 (considering sets of partitions as a single partition) that these heavier partitions will attract all future items and their topics, w.h.p. The only interesting case is when the relative loads of all partitions are close to each other, say within an $\epsilon$ fraction. However, in this case, we can apply Lemma 4 to *pairs* of partitions, to obtain that some partition will gain an monopoly.

**Lemma 5.** *Given a single cluster instance in $\mathsf{HC}(r, \ell, p, q)$ with $p \geq 2 \log r / r$ and $q = 0$ to be split across $k$ partitions, the greedy algorithm with no balancing constraints will recover the cluster onto a single partition w.h.p.*

*Proof.* Let us now fix two bins $A$ and $B$. Notice that the argument of Lemma 4 applies, up to the point where we compute $\Pr[A > B]$ in Equation 1. Here, we have to condition on either $A$ or $B$ having the maximum number of intersections, i.e., replacing

$$\Pr[\#intersections = j] = \binom{d}{j} (\sigma_t/r)^j (1 - \sigma_t/r)^{d-j}$$

with

$$\Pr[\#intersections = j \mid A \text{ or } B \text{ in argmax}].$$

Notice that the coupling still works for $j \geq 2$. Therefore, it is sufficient to show that

$$\Pr[\#intersections \in \{0, 1\}] \geq$$
$$\Pr[\#intersections \in \{0, 1\} \mid A \text{ or } B \text{ in argmax}].$$

This holds since the event ($A$ or $B$ in argmax) implies that the intersection is less likely to be empty or of size 1. Therefore, the argument reduces to the two bin case. □

**Speed of Convergence.** Note that, by Chernoff bounds, once one of the partitions acquires a constant fraction more topics from the single cluster than the other partitions, it will acquire all future topics w.h.p. By Lemma 4, it either holds that one of the partitions dominates before time $t_0 = r / \log r$, or that we can couple greedy with a Polya urn process with $\gamma > 1$ after this time. The only remaining piece of the puzzle, before we consider the multi-cluster case, is *how fast* the Polya urn process converges to a configuration where some partition contains a constant fraction more topics than the others.

This question is addressed by Drinea et al. [2], which prove the following two facts about the two-bin and $k$-bin case, respectively. We state this as a single result below, combining Theorems 2.1, 2.4, and Lemma 4 from the aforementioned paper. A system of two bins is said to $\epsilon_0$-separate if one of the bins acquires a $1/2 + \epsilon_0$ fraction of the balls. A bin $B_0$ is all-but-$\delta$ dominant if $B_0$ contains at least a $1 - \delta$ fraction of the balls thrown.

**Theorem 2** (Speed of Convergence [2]). *The following hold.*

1. *Consider a Polya urn process with $\gamma > 1$, and two bins, in an arbitrary initial state with at least one ball each. Then there exist constants $\epsilon_0$ and $\lambda > 0$ such that, after $n$ steps, the probability that the two bins fail to $\epsilon_0$ separate is at most $O(n^{-\lambda})$.*

2. *Consider a Polya urn process with $\gamma > 1$, and two bins. Assume that, initially, there are $n_0$ balls in the system, and that bin $B_0$ has an $\epsilon_0$ advantage. We throw balls until $B_0$ is all-but-$\delta$ dominant, for some $\delta > 0$. Then, with probability $1 - e^{\Omega(n_0)}$, $B_0$ is all-but-$\delta$ dominant when the system has $2^{x+z} n_0$ balls, where $x = \frac{\log(0.4/\epsilon_0)}{\log(1+(\gamma-1)/(5+4(\gamma-1)))}$, and $z = \frac{\log(0.1/\delta)}{2\gamma/(\gamma+1)}$.*

3. *Suppose that when $n$ balls are thrown into a pair of bins, the probability that neither is all-but-$\delta$ dominant is upper bounded by $p(n, \delta)$, non-increasing in $n$. Then when $1 + kn/2$ balls are thrown into $k$ bins, the probability that none is all-but-$\lambda$ dominant is at most $\binom{k}{2} p(n, \delta)$, for $\lambda = \frac{\delta}{\delta + (1-\delta)(k-1)}$.*

**Convergence argument.** We can apply the previous result to bound the convergence time of the algorithm as follows.

**Theorem 3.** *Given a hidden co-cluster graph in* $\mathsf{HC}(n, r, \ell, p, q)$*, with parameters* $p \geq 2\log r / r$*,* $q = 0$*, and a single hidden cluster, i.e.,* $\ell = 1$*, to be split across* $k$ *partitions, the following holds. There exists a partition* $j$ *such that, after* $2r / \log r$ *items have been observed, each additional generated item is assigned to partition* $j$*, w.h.p.*

## 1.6 Final Step: The General Case

We now complete the proof of Theorem 1 in the general case with $\ell \geq 2$ clusters and $q > 0$. We proceed in three steps. We first show the recovery claim for general $\ell \geq 2$, but $q = 0$ and no balance constraints, then extend it for any $q \leq \log r / (rk)$, and finally show that the balance constraints are practically never violated for this type of input.

**Generalizing to** $\ell \geq 2$**.** A first observation is that, even if $\ell \geq 2$, the topics must be disjoint across clusters if $q = 0$. Also, since we assume no balance constraints, the clusters and their respective topics are independent. The assignment problem for clusters then reduces to throwing $\ell$ balls (the *clusters*) into $k$ bins (the *partitions*). We use concentration bounds on the result bin loads to understand the maximum number of clusters per partition, which in turn bounds the maximum load.

**Lemma 6.** *Assume a clustered bipartite graph* $G$ *with parameters* $\ell \geq k\log k$*,* $p \geq 2\log r / r$*, and* $q = 0$*, to be split onto* $k$ *partitions with no balance constraints. Then, w.h.p., greedy ensures balanced recovery of* $G$*. Moreover, the maximum number of topics per partition is upper bounded by* $(1 + \beta)r\ell / k$*, w.h.p., where* $\beta < 1$ *is a constant.*

*Proof.* Notice that, since the clusters are disjoint and $q = 0$, their corresponding topics must be disjoint. Also, since there is no balance constraint, the clusters and their respective topics are independent. Fix an arbitrary cluster $C_i$. Let $t_i$ be the first time in the execution when we have observed $2r / \log r$ items from $C_i$. By Theorem 3, after time $t_i$ there exists a partition $P_j$ such that all future items associated to this hidden cluster will be assigned to $P_j$, w.h.p. Also, note that, by Lemma 2, the expected number of topics from this cluster that may have been assigned to other partitions by time $t_i$ is at most $r(1 - 1/e^2)$, which implies that at most $8m/9$ total topics may have been assigned to other partitions by this time, w.h.p.

To examine the maximum partition load, we model this process as a balls-into-bins game in which $\ell = k\log k$ balls (the clusters) are distributed randomly across $k$ bins (the partitions). The expected distribution per bin is of $\ell / k$ clusters, and, by Chernoff bounds, the maximum load per bin is $(1 + \alpha)\ell / k$, with high probability in $k$, where $0 < \alpha < 1$ is a small constant. This means that a partition may receive number of topics of $(1 + \alpha)r\ell / k$ from the clusters assigned to it. To upper bound the extra load due to duplicates, first recall that at most $8m/9$ total topics from each cluster may be duplicated, w.h.p. In total, since clusters are distinct, we obtain that $8r\ell / 9$ total topics will be duplicated, w.h.p. Since these duplicates are distributed uniformly at random, a partition may receive an extra load of $(1 + \alpha)8r\ell / 9k$ topics, w.h.p. Choosing small $\alpha$, we get that the maximum load per partition is bounded by $(1 + \alpha)r\ell / k + (1 + \alpha)8r\ell / 9k \leq 1.9r\ell / k$. It is interesting to contrast this to the factor obtained by random assignment of items to partitions. $\square$

**Generalizing to** $q > 0$**.** The next step is to show that, as long as $q < \log r / (rk)$, the greedy process is not adversely affected by the existence of out-of-cluster (noise) topics, since out-of-cluster topics have a very low probability of changing the algorithm's assignment decisions.

**Lemma 7.** *Given* $q < \log r / (rk)$*, then w.h.p. the greedy process can be coupled with a greedy process on the same input with* $q = 0$*, where* $r / \log r$ *topics have been observed for each cluster of topics.*

*Proof.* We couple the two processes in the following way. We consider a hidden cluster input $G$ built with $q = 0$, and a copy of that input $G'$ where $q = \log r / (rk)$, running the algorithm in parallel on the two graphs. Notice that we can view this process on an item-by-item basis, where in the $q = 0$ copy the algorithm gets presented with an item $\tau$, while in $G'$ the algorithm gets presented with a

variant $\tau'$ of $\tau$ from the same home cluster, which also has out-of-cluster topics, chosen uniformly at random from an arbitrary set $Q_h$ of at most $r/2$ topics.

The key question is whether the greedy assignments are the same for items $\tau$ and $\tau'$. We prove that this is indeed the case, with high probability. In particular, we need to show that, w.h.p., the outside topics are not enough to change the decision based on the intersection argmax.

Given an item $\tau'$ in $G'$ which belongs to cluster $C_i$, notice that, by Lemma 2, it has at least $3 \log r/2$ distinct topics in $C_i$, w.h.p. Let $t_i$ be the first time when at least $r/\log r$ items from $C_i$ have been observed. After time $t_i$, using Chernoff bounds and the pigeonhole principle, the size of the intersection of $\tau$ with one of the $k$ partitions must be of at least $(1 - \alpha)(1 - 1/e)3 \log r/2k$, w.h.p., where $\alpha > 0$ is a constant.

We now bound the number of topics that $\tau$ has outside $C_i$. Since $q < \log r/rk$, it follows that $\tau$ may have at most $(1 + \beta) \log r/k$ topics outside $C_i$, w.h.p., where $\beta$ is a constant. For small $\alpha$ and $\beta$, we get that the number of home cluster topics of $\tau$ exceeds the number of outside topics, w.h.p. In turn, this implies that the two random processes can be coupled for each cluster starting with time $t_i$, as claimed. $\qquad\square$

We can combine Lemma 7 and Theorem 3 to obtain that greedy converges after $2r/\log r$ items have been observed out of each hidden cluster.

**The Capacity Constraint.** Finally, we extend the argument to show that the partition capacity constraints do not cause the algorithm to change its decisions, with high probability. The proof follows by noticing that the load distributions are balanced across servers as the algorithm progresses, as items are either distributed randomly (before convergence), or to specific partitions chosen uniformly at random (after convergence).

**Lemma 8.** *On a hidden co-cluster input, greedy without capacity constraints can be coupled with a version of the algorithm with a constant capacity constraint, w.h.p.*

*Proof.* We can model the assignment process as follows: during the execution, each of the $\ell$ clusters has its items assigned randomly (at the beginning of the execution), then converges to assigning items to a single server. If we regard this from the point of view of each partition $i$ at some time $t$, there is a contribution $R_i$ of topics which comes from items in clusters that are still randomly assigned at $t$, and a contribution $F_i$ of topics coming from items in clusters that have converged. Notice that both these contributions are *balanced across partitions*: each partition has the same probability of being assigned a random cluster; also, since clusters are assigned independently and $\ell \geq k \log k$, the weight coming from converged clusters is also balanced across partitions. Using concentration bounds for each contribution in turn, it follows that the maximally loaded partition is at most a constant fraction more loaded then the minimally loaded one, w.h.p. $\qquad\square$

**Final Argument.** Putting together Lemmas 6, 7 and 8, we obtain that greedy ensures balanced recovery for general hidden cluster inputs in $\mathsf{HC}(n, r, \ell, p, q)$, for parameter values $\ell \geq k \log k$, $p \geq 2 \log r/r$, and $q \leq \log r/(rk)$. This completes the proof of Theorem 1.

Moreover, the fact that each cluster is recovered can be used to bound the maximum load of a partition. More precisely, by careful accounting of the cost incurred, we obtain that the maximum load is $2.4r\ell/k$, with high probability, where the extra cost comes from initial random assignments, and from the imperfect balancing of clusters between partitions.

## 2    Experimental Validation

**Synthetic Co-Cluster Inputs.** We also considered generated hidden co-cluster inputs. In particular, we generated hidden co-cluster graphs for various values of parameters $r, \ell, p, q$, and $m = r\ell$. We focus on two measures. The first is *recall*, which is defined as follows: for each cluster, we consider the partition that gets the *highest fraction* of topics from this cluster. We then average these fractions for all clusters, to obtain the recall. The second measure is the *maximum partition size*, i.e., the maximum number of topics on a partition after $m \log m$ items have been observed, normalized by $m/k$, which is a lower bound on the optimum. We expect these two measures to

(a) Testing the sufficient condition      (b) Experiments for non-uniform sources.

Figure 1: Experiments for hidden cluster bipartite graphs. The dotted line upper bounds the analytic recovery threshold.

be correlated, however neither one in isolation would be sufficient to ensure that greedy provides balanced recovery.

When generating the random inputs, we select a random home cluster, then subscribe to topics from the home cluster with probability $p$. When subscribing to topics from outside the home cluster, we pick *every* topic from outside the cluster independently with probability $q$ (so the noise set $Q$ contains all topics).

**Testing the Sufficient Conditions.** Our first experiment, presented in Figure 1a, fixes the value of $p$ to $2 \log r / r$, and increases the value of $q$ from $p/(10\ell)$ (below the analytic recovery threshold) to $8p/\ell$ (above the recovery threshold). The dotted line represents an upper bound on the recovery threshold $q = p/(4\ell)$. The experiment shown is for $r = 64, \ell = 64$, and $k = 20$. The results are stable for variations of these parameters.

The experiments validate the analysis, as, below the chosen threshold, we obtain both recall over $90\%$, and partition size within two of optimal. We note that the threshold value we chose is actually *higher* than the value $q = \log r / (rk)$ required for the analysis.

**Testing the Sufficient Conditions.** Our first experiment, presented in Figure 1a, fixes the value of $p$ to $2 \log r / r$, and increases the value of $q$ from $p/(10\ell)$ (below the analytic recovery threshold) to $8p/\ell$ (above the recovery threshold). The dotted line represents an upper bound on the recovery threshold $q = p/(4\ell)$. The experiment shown is for $r = 64, \ell = 64$, and $k = 20$. The results are stable for variations of these parameters.

The experiments validate the analysis, as, below the chosen threshold, we obtain both recall over $90\%$, and partition size within two of optimal. We note that the threshold value we chose is actually *higher* than the value $q = \log r / (rk)$ required for the analysis.

**Non-Uniform Clusters.** We repeated the experiment choosing home clusters with *non-uniform probability*. In particular, we select a small set of clusters which have significantly more probability weight than the others. The experimental results are practically identical to the ones in Figure 1a, and therefore omitted. These empirical results suggest that non-uniform cluster probabilities do not affect the algorithm's behavior.

**Non-Uniform Topics.** Finally, in Figure 1b, we analyze the algorithm's behavior if topics have non-uniform probability weights. More precisely, we pick a small set of topics in each cluster which have disproportionately high weight. (In the experiment shown, four sources out of $64$ have .1 probability of being chosen.) We observe that this affects the performance of the algorithm, as recall drops at a higher rate with increasing $q$.

The intuitive reason for this behavior is that the initial miss-classifications, before the algorithm converges, have a high impact on recall: topics with high probability weight will be duplicated on all partitions, and therefore their are no longer useful when making assignment decisions.