[Reviews · NeurIPS 2015]

Submitted by Assigned_Reviewer_1

This paper proposes a simple greedy strategy for the min-max hypergraph partitioning problem under the streaming model. This paper defines a stochastic model for generating the input hypergraph and show the sufficient conditions for the asymptotical recovery of the hidden clusters for the hypergraphs generated using this model.

I think the paper studies an interesting and important problem. The proposed stochastic model for graph generation seems interesting and novel. I find it interesting to see that the min-max graph partitioning problem can be almost recovered under certain conditions of the graph.

However, I am mostly concerned about how realistic the stochastic model for generating the graph is: does the hidden co-clustering assumption about how the data is generated realistic in practice? I hope to see more justification of why this model is natural for real-world applications.

I think the authors should also cite a recent but very relevant paper below, which essentially considers the same problem of min-max graph partitioning and proposes an offline greedy algorithm very similar to the one proposed in this work. The offline greedy algorithm is claimed to run with time complexity of O(k|E|) , where k is the number of parts in the resulting partitioning and |E| is the number of edges in the bipartite graph. I am curious to see how the streaming greedy algorithm performs relative to the offline greedy algorithm empirically.

Graph Partitioning via Parallel Submodular Approximation to Accelerate Distributed Machine Learning Mu Li, Dave G. Andersen, Alexander J. Smola, 2015

http://arxiv.org/pdf/1505.04636.pdf

Minor comments:

Line 168-171: It seems that this part of paper is trying to claim that the random partitioning of the graph can yield bad performance, but all it shows is that the random partitioning gives a cost of (1-1/e)m, whereas it is unclear what the global optimal is.

Description of Alg 1 (Line 216-225): The notations used in Alg 1 are a bit confusing. At the beginning of Alg 1, S_i's are defined to be the assignment of the items, which makes me think that S_i should denote the set of items in block i. However, it seems that S_i is used to refer to the set of topics covered by the items in block i in the description of the algorithm. I find it quite confusing.

Summary: This paper proposes a simple greedy strategy for the min-max hypergraph partitioning problem under the streaming model. This paper defines a stochastic model for generating the input hypergraph and show the sufficient conditions for the asymptotical recovery of the hidden clusters.

Submitted by Assigned_Reviewer_2

The paper proposes a new greedy method for streaming hypergraph clustering, with the goal of minimizing the maximum number of topics covered by a partitioning. The proposed algorithm is simple, and the authors prove the algorithm provably performs well under a balance assumption of the data, asymptotically uncovering the hidden partitioning. The algorithm is then compared against other simple algorithms on real data sets, which it outperforms in terms of normalized maximum load.

It is noted that the new techniques could not be compared against batch methods on real data sets due to their slow computational time. The experiment could be performed on smaller sets (or simply small subsets of the data sets used), and may be interesting (even though the new method cannot necessarily be expected to perform as well as batch techniques, it would be interesting to see what the gap is).

Small typo: Double "the" on line 319.
Summary: The paper proposes a new greedy method for streaming hypergraph clustering, with the goal of minimizing the maximum number of topics covered by a partitioning, proving that it performs well under balance assumptions and demonstrating its efficacy in practice. This problem is typically studies in the batch setting, this makes a natural step and studies it in the streaming model, and the new method seems to work well.

Submitted by Assigned_Reviewer_3

Overall, I liked the paper. The paper addresses an important problem, and has a nice balance of theoretical results and empirical ones. Some comments:

1) The authors should mention that the problem they consider is a special case of submodular load balancing, studied in 'Submodular approximation: sampling-based algorithms and lower bounds' (Svitkina & Fleicher).

2) The result about arbitrary partitions (Proposition 2) was known for general submodular functions (see, again the above reference). 3) Another related paper -- Graph Partitioning via Parallel Submodular Approximation to Accelerate Distributed Machine Learning (Mu Li et al)
Summary: This paper studies the problem of Min-Max hyper-graph partitioning in a streaming setting. The authors observe that this problem is an instance of min-max subodular data partitioning, a problem also called the submodular load balancing problem. The authors show theoretical results, and demonstrate the algorithm on real world data.

Author Feedback
Author rebuttal: We thank the reviewers for their comments. Answers below:

Reviewer 1:

- The Stochastic Model:
The intuition behind the model is that items, e.g. user interests, are drawn from a hidden mixture distribution across topics. As such, our model is an instantiation of the classic mixture model [Kleinberg-Sandler, STOC04], widely used in collaborative filtering. The choice of distribution follows the line of research on the planted partition / stochastic block model. Our model is a natural generalization of the stochastic block graph model to hypergraphs.
We do not claim that our model exactly matches real data. However, we find it interesting that an algorithm with the recovery property in the hidden cluster model consistently outperforms algorithms without this property, on real data. (Our framework can also show that other partitioning strategies considered, e.g. proportional or random, do not have the recovery property, as they couple to diverging Polya urn processes.)
Our work is the first to provide analytic guarantees for streaming hypergraph partitioning. We hope that it will spark further research, in particular on generalizing the stochastic input model.

- The Li, Andersen, Smola [LAS15] paper:

We were unaware of this work at submission time, given its timing. We plan to include a complete comparison in the next version of our paper, and provide a brief overview below.

- The two papers are complementary in terms of contribution. We consider online arrivals, motivated by streaming query processing platforms. [LAS15] considers an offline model, where the entire input is available to the algorithm, and a key component is a procedure for optimizing the order of examining vertices, and its efficient implementation. By contrast, we show that, even for random arrival order, our algorithm performs well.

- We compared our algorithm with Parsa on a subset of inputs. On a bipartite graph (news20), compared to random, greedy gives a 20% improvement in memory, 60% improvement in total traffic, and 71% improvement in max traffic. On a social graph (livejournal), greedy gives a 67% improvement in memory, and ~100% improvement in (max and total) traffic.
These results put it at around 50% of Parsa's performance in terms of these parameters, which can be justified since we stream the input. Running time is less than that of Parsa (similar to PowerGraph).

- Performance of random assignment:

Consider an input where m topics form k disjoint clusters, and each item only subscribes to a single cluster. The optimal way to distribute onto k partitions would be to place each cluster onto one partition, with cost m / k. By the Lemma, random partitioning has cost ~2 m / 3. Our point is that the competitive ratio is linear in k, almost as bad as placing all items onto the same partition.
We will clarify this part of the discussion.

- We will reword the description to clarify that S_i's are sets of topics

Reviewer 2:
- Comparison to offline techniques:

On bipartite data sets, which are our focus, hMETIS gives better cuts (up to 2x better), but very unbalanced partitions (>3x difference between min and max size, on the podcast data), since it is designed to optimize for cut size. Label propagation and spectral methods had high running times and gave inferior results on sparsified inputs (even on hidden cluster synthetic inputs).

Comparison with Parsa (see above), and implicitly Zoltan and PaToH, on bipartite inputs suggests that performance is between 20-60% relative to these offline methods.

On social graphs, performance is usually 20-50% of that of offline techniques such as METIS.

Reviewer 3:

- 1) and 2): Indeed, we are considering a specific instance of submodular load balancing. We will clarify the connection.
- 3) : Please see response to Reviewer 1

Reviewer 4:

- Our results are a strict generalization of [19], as we consider partitioning of hypergraphs. One key technical distinction is in the reduction to Polya urn processes: while for graphs the coupling is straightforward, in the case of hypergraphs the coupling hinges on relating the number of items and the number of topics on each partition.

Reviewer 5:
- We erroneously submitted a deprecated version of the additional material, and were unable to replace it after the deadline. The complete argument is available in the technical report version of our paper. (Available online, but not anonymized.) This argument is a significant part of our technical contribution.

Reviewer 6:
- Indeed, an item subscribing to most topics forces up the max partition cost. We have not found such items (users) in the data sets we analyzed. In practice, we believe these items would get special treatment, i.e. be split up and distributed across partitions. Extremely popular topics do exist (<20%), and are duplicated on each partition.